# MindLoc: A Secure Brain-Based System for Object Localization

## Abstract

Object localization tasks aim to accurately locate and identify specified target objects within images, representing a core challenge in the field of computer vision. Traditional object localization systems primarily rely on intermediary modalities such as text descriptions, speech, or visual cues to interpret human intent. However, these modalities only provide indirect expressions of human intent, limiting the efficiency of information transmission. This is particularly evident when detailed descriptions of texture and spatial information are required, resulting in higher interaction costs. While existing brain-based object localization systems offer the potential for directly interpreting human intent, their localization accuracy still lags behind traditional text-based systems. Additionally, the high cost of data collection, limited diversity of participants, and significant individual cognitive differences make it challenging to train subject-independent models, thereby constraining the development of brain-based object localization systems. To address the challenges, we propose MindLoc, a lightweight, cross-subject brain-based object localization model. MindLoc can rapidly and accurately locate target objects in complex images by directly analyzing fMRI signals, combining the precision of traditional localization systems with the convenience of brain-based systems. Additionally, we are the first to introduce encryption technology for the privacy protection of brain data, significantly reducing the psychological burden on participants, which provides a foundation for increasing participant diversity in future studies. Experimental results demonstrate that MindLoc has achieved new state-of-the-art performance in brain-based object localization tasks, showcasing significant advantages in both accuracy and convenience. Our code is available at https://mindloc-sys.github.io/.

## 1 Introduction

AI research typically relies on intermediary modalities, such as speech Shi et al. (2022), text Radford et al. (2021b), and images Yang et al. (2024), to understand human intent. However, these modalities are merely indirect channels of communication, serving as abstractions of consciousness, whereas neural signals represent consciousness itself, with an information entropy far higher than that of other modalities. Therefore, directly interpreting neural signals holds great potential.

Finding targets in a complex environment can be a challenging task. Traditional text-based object localization systems Carion et al. (2020); Liu et al. (2023) have been highly refined, while brain-based systems still lag far behind in localization capability Xia et al. (2024). However, traditional localization systems have critical limitations. For instance, conveying details like texture and spatial information through text descriptions comes with a high cost, because the user not only has to carefully organize their thoughts but also manually input the information. Our system combines the advantages of both, with localization capabilities comparable to those of traditional localization systems and the advantages of convenient, easy-to-describe requirements of brain-based target localization systems. Think about the process of searching for targets: the general appearance of the object usually comes to mind. Now, imagine a pair of glasses that can capture the current scene and, based on brain responses, quickly locate the desired object and highlight it on the display. Wouldn't that be remarkable? Its applications are not limited to this; it also holds significant value for specific target detection in the military field and industrial scenarios. Based on this, we propose MindLoc, a

Figure 1: The main ability of our model. Our system takes brain data and a visual scene as input, allowing for the localization of the desired object. Traditional models, relying on text descriptions, might identify a green apple because it stands out more. In contrast, our system incorporates brain-wave information, enabling it to locate the specific red apple we want. CSs refer to the processing systems, where CS1 possesses both a public and private key, while CS2 only holds the public key.

lightweight, cross-subject object localization system. As shown in Fig. 1, our system can accurately locate objects that users associate with, using only brain signals, even in complex images.

Additionally, a significant obstacle to the development of brain-computer interface (BCI) technology is the high cost of data collection and the limited diversity of participants. Moreover, the considerable variation in individual thought makes it challenging to train a universal, subject-independent model. Enhancing subject robustness has thus become an important research direction. From experience in other fields, scaling up Wang et al. (2020) appears to be the key to breakthroughs. For instance, ChatGPT Radford et al. (2019); Brown et al. (2020) in the area of natural language processing, and CLIP Radford et al. (2021b) in computer vision are both trained on large datasets. Based on this, we introduced encryption technology for the privacy protection of brain signals, significantly reducing the psychological burden on participants, which lays the foundation for increasing participant diversity in future studies.

Our main contributions are as follows:

- We propose MindLoc, a lightweight, cross-subject brain-based object localization model, which can rapidly and accurately locate target objects in complex images by directly analyzing fMRI signals.
- We are the first to introduce encryption technology for the privacy protection of brain data, significantly reducing the psychological burden on participants, which provides a foundation for increasing participant diversity in future studies.
- Our system has achieved new state-of-the-art (SOTA) performance in brain-based localization tasks, combining the precision of traditional localization systems with the convenience of brain-based systems.

## 2 RELATED WORK

### 2.1 BRAIN SIGNAL COMPREHENSION

Recently, generative visual models based on brain signals Lin et al. (2022); Ozcelik & VanRullen (2023); Scotti et al. (2024); Takagi & Nishimoto (2023b); Xia et al. have demonstrated exceptional performance in decoding visual stimuli from brain responses. Typically, these methods map brain responses captured through functional magnetic resonance imaging (fMRI) to common modalities suitable for input into pre-trained visual-language models Karras et al. (2020); Rombach et al. (2022); Xu et al. (2023), facilitating subsequent image reconstruction. For example, Lin et al. (2022) projects fMRI data into the CLIP Radford et al. (2021a) space, which embeds images and captions, and subsequently utilizes the Lafite model, adjusted using an unconditional StyleGAN2 Karras et al.

(2020) framework, to perform image reconstruction. Building on this foundation, Scotti et al. (2024) establishes a high-level (semantic) pathway for mapping fMRI data to the CLIP ViT-L/14 image space and a low-level (perceptual) pathway for mapping fMRI data to the image embedding space of a Variational Autoencoder (VAE). However, Scotti et al. (2024) is limited to natural scenes in the MS-COCO dataset; for other image distributions, additional data collection and specialized generative models are required. Takagi & Nishimoto (2023b) utilized ridge regression to link fMRI signals with CLIP text embeddings and the diffusion model of Stable Diffusion (SD) Rombach et al. (2022), enabling a simple linear mapping from fMRI to the latent representation of the latent diffusion model (LDM), thereby reducing the computational cost of the diffusion model. Xia et al. proposed the "DREAM" method, which extracts semantic, depth, and color information to reconstruct images using depth-color conditional SD. This approach Han et al. (2024); Takagi & Nishimoto (2023a) differs from previous methods that relied solely on text embeddings to obtain explicit descriptions of visual stimuli. Ozcelik & VanRullen (2023) introduced the Brain-Diffuser, a two-step generative framework that first generates images with foundational features and overall layout using the VDVAE model, followed by employing a state-of-the-art generative latent diffusion model for subsequent image reconstruction.

## 2.2 IMAGE LOCALIZATION

Carion et al. (2020) introduced the DETR detection model, which was subsequently improved by Chen Chen et al. (2023b); Dai et al. (2021); Gao et al. (2021); Jia et al. (2023); Meng et al. (2021); Wang et al. (2022); Zhu et al. (2020) through various enhancements, including the Group DETR Chen et al. (2023b) and Deformable DETR models Zhu et al. (2020). However, these models primarily operate within a closed set of predefined categories, making it challenging to extend their application to new categories. This limitation has prompted research into open-set object detection, which utilizes existing bounding box annotations for training and employs language generalization to achieve the detection of arbitrary categories. OV-DETR Zareian et al. (2021) employs image and text embeddings encoded by the CLIP model as query requests to decode specific class boxes within the DETR framework Carion et al. (2020). ViLD Gu et al. (2021) extracts knowledge from a CLIP teacher model and transfers it to an R-CNN-like detector, allowing the learned region embeddings of the detector to incorporate text and images inferred from the teacher model. GLIP Gao et al. (2024) reformulates the object detection task as a foundational problem and incorporates additional foundational data to achieve semantic alignment at various levels. Experiments demonstrate that this model can achieve stronger performance on fully supervised detection benchmarks. DetCLIP Yao et al. (2022), on the other hand, leverages a generated concept dictionary to expand its knowledge base for large-scale image-caption datasets, effectively enhancing the model's generalization capabilities. However, prior work has predominantly focused on traditional modalities for localization tasks, while recent efforts have begun to explore direct localization through EEG signals, albeit with significantly lower accuracy compared to traditional models.

## 3 METHOD

As shown in Fig. 2, we introduce the MindLoc model, which can locate objects solely based on brain signals. First, fMRI signals are transformed into embeddings through an extractor, as detailed in Sec. 3.1. The security fusion module generates combined features and transfers the category information to the localization module, as explained in Sec. 3.2. The localization module reads the background image and ultimately outputs the localization result, which will be detailed in Sec. 3.3.

## 3.1 MULTIMODAL ALIGNMENT

We design an extractor, which learns to map flattened spatial patterns of fMRI activity across voxels (3-dimensional cubes of cortical tissue) to the image embedding latent space of a pretrained CLIP model. The structure simplifies and optimizes to meet the requirements of being lightweight, with the number of parameters reduced to only 1/10 of Scotti et al. (2024). The network incorporates convolutional concepts and is jointly trained on the NSD Allen et al. (2022) and GOD Horikawa & Kamitani (2017) datasets. We employ multi-task learning to align the fMRI modality, image modality, and dual-level text modality within the CLIP feature space, achieving both sentence-level and word-level alignment.

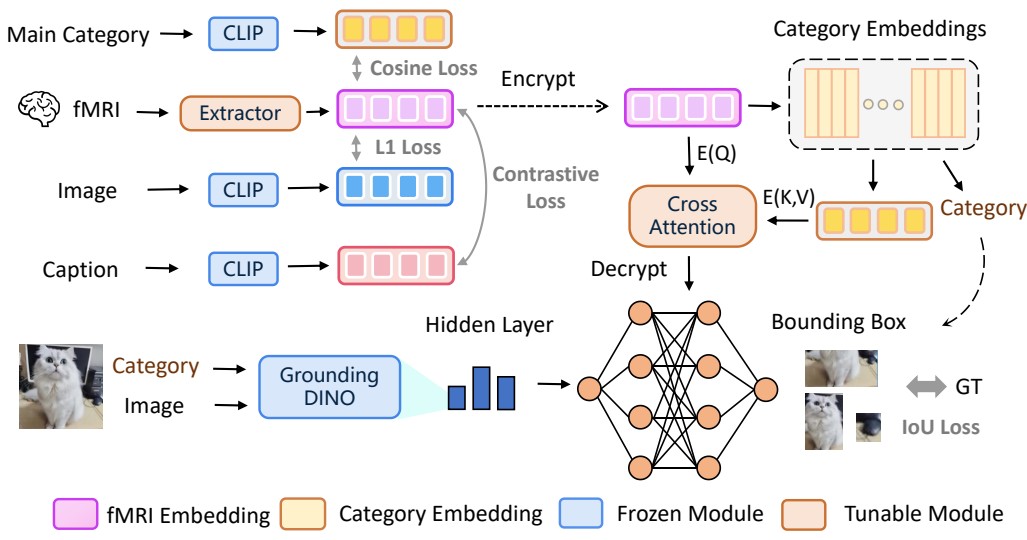

Figure 2: The overview of the MindLoc. The model is composed of three main components: a feature extraction module, a search module under encrypted conditions which also provides feature fusion capabilities, and an object localization module. During the inference phase, the input consists solely of fMRI signals and the current visual frame.

**fMRI-Cat Loss**   Let the encoded result of the text for each category be $f_{cat}^c$, where $c \in \mathbb{R}^C$, with $C$ representing the number of categories. For each image in the dataset, the encoded result is $f_{img}^i$, where $i \in \mathbb{R}^N$, with $N$ representing the number of images. We calculate the matrix $SIM \in \mathbb{R}^{N \times C}$ through:

$$sim(i, c) = \frac{f_{cat}^c \cdot f_{img}^i}{\left\| f_{cat}^c \right\|_2 \cdot \left\| f_{img}^i \right\|_2} \in (0, 1) \tag{1}$$

Taking the maximum value across columns, $\hat{c} = \arg\max(SIM) \in \mathbb{R}^N$, then $\hat{c}_i$ is designated as the main category for the i-th image. Let the result of the fMRI information extracted by the extractor be $f_{fMRI}^i$, then the alignment loss between fMRI and the main category can be expressed as:

$$L_{fMRI-cat} = \frac{f_{cat}^{\hat{c}_i} \cdot f_{fMRI}^i}{\left\| f_{cat}^{\hat{c}_i} \right\|_2 \cdot \left\| f_{img}^i \right\|_2} \tag{2}$$

**fMRI-Img Loss**   The fMRI features and image features are aligned in L1 space. Define $\tau$ as a temperature ratio, which is a hyperparameter that weights the degree of fMRI-image alignment. Then the $L_{fMRI-img}$ can be expressed as:

$$L_{fMRI-img} = \mathcal{F}\left( \left\| f_{fMRI}^i - f_{img}^i \right\|_1 \cdot \tau \right) \tag{3}$$

where $\mathcal{F}$ is a mapping function. $L_{fMRI-img}$ focus the model on the information concerning image, restoring the features such as color.

**fMRI-Cap Loss**   Contrastive learning is an effective representation learning method that learns representations across multimodal data by maximizing the cosine similarity of positive sample pairs and minimizing the similarity of negative sample pairs. Previous research suggests that combining contrastive learning with neural data can yield significant benefits Défossez et al.; Schneider et al. (2023). CLIP is an example of a multimodal contrastive model that maps images and text captions to a shared embedding space. MindLoc is designed to incorporate fMRI as an additional modality into the embedding space of a pretrained CLIP model, while keeping the CLIP image space fixed, similar to the approach used in locked-image text tuning (LiT). We utilize the CLIP loss as our contrastive

---

**Algorithm 1** $\text{SM}(E(a), E(b)) \rightarrow E(a \times b)$

---

1: **Input:** $CS_2$ has $E(a)$ and $E(b)$; $CS_1$ has $sk$
2: **Output:** $CS_2$ obtains the product $E(a \times b)$
3: $CS_2$ generates two random numbers $r_a, r_b \in \mathbb{Z}_N$
4: $CS_2$ calculates $a' \leftarrow E(a) \times E(r_a)$, $b' \leftarrow E(b) \times E(r_b)$
5: $CS_2$ sends $a', b'$ to $CS_1$
6: $CS_1$ decrypts $h_a \leftarrow D(a')$ and $h_b \leftarrow D(b')$
7: $CS_1$ calculates $h' \leftarrow h_a \times h_b \mod N$
8: $CS_1$ encrypts $h' \leftarrow E(h')$ with $pk$
9: $CS_1$ sends $h'$ to $CS_2$
10: $CS_2$ receives $h'$ from $CS_1$
11: $CS_2$ calculates $s \leftarrow h' \times E(a)^{N-r_b}$ and $s' \leftarrow s \times E(b)^{N-r_a}$
12: $CS_2$ calculates $E(a \times b) \leftarrow s' \times E(r_a \times r_b)^{N-1}$

---

objective. Let the embedding representation of the caption after processing by CLIP be denoted as $f_{cap}$. Then, $L_{fMRI-cap}$ can be expressed as:

$$S = \frac{f_{fMRI} \cdot f_{cap}^T}{\tau} \tag{4}$$

$$L_{fMRI-cap} = -\frac{1}{N} \cdot \sum_{i=1}^{N}[\lambda \cdot log(\frac{e^{S_{ii}}}{\sum_j e^{S_{ij}}}) + (1 - \lambda) \cdot log(\frac{e^{S_{ii}}}{\sum_j e^{S_{ji}}})] \tag{5}$$

where $\tau$ is a temperature hyperparameter, and $\lambda$ controls the degree of contrastive learning in two directions.

Then the loss of the extract can be expressed as:

$$L_{total} = \lambda_1 \cdot L_{fMRI-cat} + \lambda_2 \cdot L_{fMRI-img} + \lambda_3 \cdot L_{fMRI-cap} \tag{6}$$

where $\lambda_1, \lambda_2, \lambda_3$ scale the contributions of different loss items.

## 3.2 SECURITY FUSION

A major barrier to collecting brain data comes from participants' resistance to others probing into their thoughts and memories Xia et al. (2024). fMRI signals contain a lot of private information, and there is no guarantee that the extracted features won't expose this information. Users do not want their thoughts and memories to be deciphered. In order to protect information security, we introduce the Paillier encryption technique Paillier (1999) to realize modal information fusion in the encrypted situation, which provides psychological assurance to participants and alleviates the barriers to collecting brain data. We divide the whole process into two systems, CS1 and CS2, where CS1 provides only local computing power and has both the public key and the private key. CS2 provides cloud server arithmetic, is responsible for more complex reasoning, but has only the public key.

The Paillier encryption scheme is a public-key cryptographic scheme whose security is based on an assumption related to the difficulty of factorisation. The Paillier encryption scheme is resistant to CPA Castagnos (2008) attacks if the decisional composite residuosity problem is hard. Its encryption and decryption are defined as follows:

Let $P = p \cdot q$, where $p$ and $q$ are two different large prime numbers of equal length. For any plaintext message $m \in Z_N$, its paillier homomorphic encryption function is denoted as $E$. The ciphertext $c$ corresponding to $m$ is defined as:

$$E(m) = (1 + P)^m \times r^P \mod P^2 \tag{7}$$

where $r \in Z_N$ is a random number and the public key is $P$.

The decryption function $D$ is defined as:

$$D(E(m)) = \frac{c^{\lambda(P)} \mod P^2 - 1}{P} \cdot \lambda(P)^{-1} \mod P \tag{8}$$

where the secret key is $\lambda(N) = \text{LCM}(p - 1, q - 1)$, the LCM is the least common multiple.

It can be easily observed from Eq. 7 that the Paillier encryption scheme exhibits the additive homomorphic property, meaning that decrypting the product of two ciphertexts is equal to the sum of their corresponding plaintexts, which can be demonstrated as:

$$\forall m_1, m_2 \in \mathbb{Z}_N \tag{9}$$

$$D(E(m_1) \cdot E(m_2) \mod P^2) = (m_1 + m_2) \mod P \tag{10}$$

Encrypting $f_{fmri}^i$ gives $E(f_{fmri}^i) \in R^D$, and encrypting $f_{cat}$ gives $E(f_{cat}) \in R^{C \times D}$. CS1 transmits $E(f_{fmri}^i)$ to CS2. For $c \in R^C$:

$$E(f_{fmri}^i \cdot f_{cat}^c) = E(\sum_{j=1}^{D} f_{fmri}^{ij} \cdot f_{cat}^{cj}) = \prod_{j=1}^{D} E(f_{fmri}^{ij} \cdot f_{cat}^{cj}) = \prod_{j=1}^{D} SM(E(f_{fmri}^{ij}), E(f_{cat}^{cj})) \tag{11}$$

where SM is the secure multiplication algorithm, which is presented as Alg. 1.

CS2 transmits the generated $E(f_{fmri}^i \cdot f_{cat})$ to CS1 for decryption and computes $f_{cat}^{\hat{c}} = \arg\max_{c \in C}(f_{fmri}^i \cdot f_{cat}^c)$. CS1 then sends $E(f_{cat}^{\hat{c}})$ back to CS2, where CS2 integrates the fMRI signals and category features through a cross-attention mechanism to obtain the fusion vector $f_{fusion} \in R^D$. For the fMRI features of the i-th frame, $f_{fmri}^i \in \mathbb{R}^D$, $f_{fusion}$ can be obtained using the following algorithm:

$$E(f_{fussion}^j) = \sqrt[\sqrt{D}]{\prod_{k=1}^{D} SM(E(f_{fmri}^{ij}), E(f_{cat}^{\hat{c}k}), E(f_{cat}^{\hat{c}k}))} \tag{12}$$

where SM also utilizes Alg. 1. The derivation of Eq. 12 is provided in the appendix.

## 3.3 LOCALIZATION MODULE

The model inputs are a large picture and the fMRI signal of the region of interest. The picture and the category signals generated by the fMRI are passed through the Grounding DINO module, and then the information generated by the hidden layer is transmitted to the fusion network, which simultaneously integrates the information from the security-fusion module. The candidate boxes of a given image are denoted as $\{Q_i\}_{i=1}^n$, where n is the number of candidate boxes for the image and $Q_i$ is composed of $(x, y, width, height)$. The ground truth boxes for the image are denoted as $\{G_i\}_{i=1}^m$, where m is the number of ground truth boxes for the image. We optimize the model's multi-object detection capability directly to improve its performance. First, we maintain a cost matrix $L_{metric}$. To ensure accurate matching, we apply the Hungarian algorithm Kuhn (2004) to calculate the total loss based on $L_{metric}$. The cost matrix $L_{metric}$ consists of two components: the classification loss $L_{class}$, which measures the difference between the predicted and true categories, and the IoU loss $L_{IoU}$, which evaluates the overlap between the predicted and ground truth boxes to improve the model's localization accuracy. By combining these two loss components, the model can simultaneously optimize both the target's category prediction and location matching.

**Classification Loss** In object localization tasks, the first priority is to ensure that the model correctly classifies the objects in the candidate boxes. To achieve this, we use the cross-entropy loss function to measure the difference between the predicted and true categories. The classification loss between the i-th candidate box and the j-th ground truth box can be expressed as:

$$L_{class}^{ij} = -\sum_{c=1}^{C} y_{j,c} \log(\hat{p}_{i,c}) \tag{13}$$

where $c$ represents the class, and $y_{j,c}$ indicates whether the j-th ground truth box belongs to class $c$, with $y_{j,c} = 1$ if it does. Similarly, $\hat{p}_{i,c}$ denotes the probability that the i-th candidate box is predicted to belong to class $c$.

**IoU Loss**    In addition to ensuring that the predicted candidate boxes match the ground truth boxes in terms of classification, it is also important to accurately localize the objects. For this, we introduce the IoU loss. The formula for calculating the IoU between the i-th candidate box $Q_i$ and the j-th ground truth box $G_j$ is:

$$L_{IoU}^{ij} = 1 - \frac{Area(Q_i \cap G_j)}{Area(Q_i \cup G_j)} \tag{14}$$

where $Area$ represents the area of the target region.

The total loss matrix is expressed as:

$$L_{metric}^{ij} = \alpha \cdot L_{class}^{ij} + \beta \cdot L_{IoU}^{ij} \tag{15}$$

where $\alpha, \beta$ scale the contributions of different loss items.

By applying the Alg. 2 provided in the appendix, the minimum total loss can be achieved. This approach ensures a globally optimal solution rather than a local optimum.

## 4    EXPERIMENT

### 4.1    DATASET

The Natural Scenes Dataset (NSD) Allen et al. (2022) is an essential resource for neuroscience and computer vision research. It consists of fMRI data from 8 human subjects who viewed 10,000 unique natural scene images. This dataset captures high-resolution brain activity during image viewing and includes detailed information about the images' features. It aids in understanding how the brain processes visual information from natural scenes and serves as a foundation for developing and evaluating visual models in computer vision.

The Generic Object Decoding (GOD) dataset Horikawa & Kamitani (2017) is a foundational resource for brain-computer interface (BCI) research. It comprises fMRI data collected by presenting images from 200 object categories of ImageNet Deng et al. (2009), which was released in the fall of 2011. The dataset includes a training set of 1,200 images from 150 categories and a test set of 50 images from independent categories. Five subjects were scanned during the study. This dataset provides valuable insights for understanding the relationship between brain activity and visual stimuli.

### 4.2    IMPLEMENT DETAILS

The extractor module is designed using convolutional principles and comprises a total of 130M trainable parameters. It was trained for 1,000 epochs on four A800 GPUs. The model architecture includes one ConvBlock Alaeddine & Jihene (2021), three ResidualBlocks Goceri (2019), three TransformerBlocks Min et al. (2022) each consisting of four layers, one Qformer Zhang et al. (2024), and several fully connected layers. For the learning rate scheduler, we use a LambdaLR Paszke et al. (2019) with a warm-up period of 100 iterations. The learning rate starts at zero and increases linearly to the maximum learning rate of 1e-4 during the warm-up. After that, it is adjusted from 1e-4 to a minimum of 1e-7 using cosine annealing over a total of 1000 iterations.

### 4.3    METRICS

In our study, we conducted a comprehensive evaluation of the performance of our proposed method using several key metrics that are crucial for understanding its effectiveness. One of the primary metrics, Accuracy (acc@m), measures the percentage of correctly labeled instances based on the Intersection over Union (IoU) Zhou et al. (2019) exceeding a specified threshold. We placed particular emphasis on the acc@0.5 metric, as it is critical for ensuring reliable localization of objects within the images Xia et al. (2024).

To assess the model's performance more thoroughly, we calculated the IoU to evaluate the degree of overlap between the predicted bounding boxes and the ground truth boxes. This calculation is essential in determining how accurately our method predicts the location of target objects. Furthermore, we classified the target objects into two distinct categories: "Salient" and "Inconspicuous".

Table 1: Localization accuracy. Text-based models refer to location systems that rely on textual input, offering high accuracy but also incurring significant interaction costs. Brain-based models, on the other hand, achieve localization through brain signals. UMBRAE-S refers to the model trained with a single subject only. Shikra-w/method provides visual grounding results using images produced by visual decoding methods. Further description can be found in the appendix.

| Method | All | | Salient | | SalientCreatures | | SalientObjects | | Inconspicuous | |
|---|---|---|---|---|---|---|---|---|---|---|
| | acc@0.5 | IoU | acc@0.5 | IoU | acc@0.5 | IoU | acc@0.5 | IoU | acc@0.5 | IoU |
| Text-based | | | | | | | | | | |
| Grounding DINO | 80.16 | 48.66 | 80.44 | 47.19 | 81.06 | 44.08 | 77.07 | 44.50 | 78.63 | 42.47 |
| Shikra-w | 51.96 | 47.22 | 62.92 | 56.44 | 66.71 | 59.34 | 58.79 | 53.27 | 38.29 | 35.71 |
| fMRI-based | | | | | | | | | | |
| Shikra-w/BrainDiffuser | 17.49 | 19.34 | 27.18 | 27.46 | 38.71 | 34.63 | 14.62 | 19.66 | 5.39 | 9.20 |
| Shikra-w/MindEye | 15.34 | 18.65 | 23.83 | 26.96 | 29.29 | 31.64 | 17.88 | 21.86 | 4.74 | 8.28 |
| Shikra-w/DREAM | 16.21 | 18.65 | 26.51 | 27.35 | 34.43 | 33.85 | 17.88 | 20.28 | 3.35 | 7.78 |
| Shikra-w/UMBRAE | 16.83 | 18.69 | 27.10 | 27.55 | 34.14 | 33.65 | 19.44 | 20.92 | 4.00 | 7.64 |
| UMBRAE-S | 13.72 | 17.56 | 21.52 | 25.14 | 26.00 | 29.06 | 16.64 | 20.88 | 4.00 | 8.08 |
| UMBRAE | 18.93 | 21.28 | 30.23 | 30.18 | 39.57 | 36.64 | 20.06 | 23.14 | 4.83 | 10.18 |
| MindLoc (Ours) | **64.13** | **67.08** | **67.18** | **67.79** | **70.11** | **68.65** | **61.94** | **62.69** | **61.79** | **63.92** |

Overall, these metrics provided a systematic framework for analyzing and highlighting the advantages of the MindLoc method, particularly in terms of its accuracy and adaptability to varying conditions and object types. Through this comprehensive evaluation, we aim to demonstrate the robustness and utility of MindLoc in advancing object localization tasks.

## 4.4 MAIN RESULTS

To evaluate our approach, we detect the queried objects and report both accuracy and Intersection over Union (IoU) metrics. The accuracy metric, denoted as "acc@m", measures the percentage of correctly labeled instances with an IoU exceeding the threshold m.

In Tab. 1, we first report the non-fMRI-based localization baselines. We present the results of Grounding DINO Liu et al. (2023), which has demonstrated excellent localization capabilities. We also provide the approximate results of Shikra-w, which uses ground truth images for visual grounding. Since there aren't enough brain-grounding baselines, we combine Shikra with images from state-of-the-art visual decoding methods, referred to as Shikra-w/method. Additionally, we compare our approach with the recent brain-grounding approach, UMBRAE. Our method significantly outperforms these baselines while being more lightweight and much faster in feature extraction.

Inspired by previous work Xia et al. (2024), we report metrics for all categories (referred to as "All"), while also drawing from neuroscience research on the brain's salience-processing systems Uddin (2015) to provide a more nuanced evaluation. Specifically, we group the eighty classes from the COCO dataset Lin et al. (2014) according to their salience: "Salient" categories, which include "Salient Creatures" (like people and animals) and "Salient Objects" (such as cars, beds, and tables), and "Inconspicuous" categories (such as backpacks, knives, and toothbrushes).

Our testing mainly focuses on the localization of single objects, as the human brain naturally possesses attention mechanisms. Treisman (1986) The experimental results demonstrate that MindLoc significantly outperforms previous methods across all settings, including "All", "Salient", "Salient Creatures", "Salient Objects", and "Inconspicuous". Its performance is also very close to that of state-of-the-art models that rely on text as an intermediary. Moreover, it is evident that previous approaches have significant shortcomings in locating inconspicuous objects, while our method addresses and overcomes these weaknesses. We visualized the results, as shown in Fig. 3.

## 4.5 ABLATION

In this section, we conduct ablation studies on the model architecture and training strategies to assess the impact of each component on MindLoc's performance. As shown in Tab. 2, we modify the fMRI

Figure 3: The visualization of the MindLoc. These images are sourced from the NSD dataset.

Table 2: Ablation Study. w/o FUSION Module indicates that only the Category is provided; w/o Multimodal Alignment refers to using a refined structure of Scotti et al. (2024) to extract features; w/o Main Category refers to the absence of main category optimization; and w/o Grounding DINO means using an alternative base localization model Chen et al. (2023a).

| Method | All | | Salient | | SalientCreatures | | SalientObjects | | Inconspicuous | |
|---|---|---|---|---|---|---|---|---|---|---|
| | acc@0.5 | IoU | acc@0.5 | IoU | acc@0.5 | IoU | acc@0.5 | IoU | acc@0.5 | IoU |
| MindLoc | 64.13 | 67.08 | 67.18 | 67.79 | 70.11 | 68.65 | 61.94 | 62.69 | 61.79 | 63.92 |
| w/o Multimodal Alignment | 63.52 | 65.89 | 66.59 | 66.38 | 69.45 | 67.69 | 61.37 | 61.77 | 62.11 | 64.21 |
| w/o FUSION Module | 60.92 | 63.72 | 63.82 | 64.4 | 66.60 | 65.21 | 58.84 | 59.55 | 58.70 | 60.72 |
| w/o Main Category | 60.97 | 63.25 | 63.92 | 63.72 | 66.67 | 64.98 | 58.91 | 59.29 | 59.62 | 61.64 |
| w/o Grounding DINO | 44.16 | 40.13 | 53.48 | 47.97 | 56.70 | 50.43 | 49.97 | 45.27 | 32.54 | 30.35 |
| w/o Contrastive Learning | 62.24 | 64.57 | 65.25 | 65.05 | 68.06 | 66.33 | 60.14 | 60.53 | 60.86 | 62.92 |

feature extractor and adjust the fusion module, where only the category information is provided in the localization module. We also test the effect of removing the main category optimization. Additionally, we explore the use of an alternative base localization model as described by Chen et al. (2023a). Furthermore, we replace contrastive learning with cosine loss to evaluate its impact on performance. Since previous work Xia et al. (2024); Lin et al. (2022); Scotti et al. (2024) has already extensively studied the effectiveness of various techniques Défossez et al. (2023), we will not delve into further discussion. Our experiments show that removing these modules leads to a decline in performance. Moreover, without the main category optimization, the drop in results under the Salient setting demonstrates the importance of the module. Additionally, the experimental results indicate that the performance of our model is tied to its fundamental models. Our model is also compatible with other localization models, paving the way for future iterations.

## 5 CONCLUSION

In this paper, we presented MindLoc, a novel lightweight and cross-subject brain-based object localization model that leverages fMRI signals to accurately identify and locate target objects within complex visual scenes. A key innovation of our approach is the integration of encryption technology to protect brain data privacy, which reduces the psychological burden on participants and supports greater diversity in future studies. Experimental results demonstrate that MindLoc achieves state-of-the-art performance in brain-based localization tasks, combining the precision of traditional systems with the convenience of brain-based approaches. This advancement highlights the potential of brain-computer interface (BCI) technologies in various applications, including assistive technologies, military target detection, and industrial automation. Given that fMRI data typically contain less signal noise compared to Electroencephalogram (EEG), our research primarily focused on fMRI. In future work, we plan to shift our efforts toward EEG to enhance real-time applications. Overall, Mind-Loc represents a significant step forward in brain-based object localization, offering high accuracy, user-friendly features, and robust privacy protections, and is expected to inspire further research and development in the field of BCI.

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

# A APPENDIX

## A.1 THE DERIVATION OF EQ. 12

For the fMRI features of the i-th frame, $f_{fmri}^i \in \mathbb{R}^D$, $f_{fusion}$ can be obtained using the following algorithm:

$$E(f_{fussion}^j) = E(\frac{f_{fmri}^{ij} \cdot f_{cat}^{\hat{c}}}{\sqrt{D}} \cdot (f_{cat}^{\hat{c}})^T) \tag{16}$$

$$E(f_{fussion}^j) = E(\frac{\sum_{k=1}^{D} f_{fmri}^{ij} \cdot f_{cat}^{\hat{c}k} \cdot f_{cat}^{\hat{c}k}}{\sqrt{D}}) \tag{17}$$

$$E(f_{fussion}^j) = \sqrt[\sqrt{D}]{E(\sum_{k=1}^{D} f_{fmri}^{ij} \cdot f_{cat}^{\hat{c}k} \cdot f_{cat}^{\hat{c}k})} \tag{18}$$

$$E(f_{fussion}^j) = \sqrt[\sqrt{D}]{\prod_{k=1}^{D} SM(E(f_{fmri}^{ij}), E(f_{cat}^{\hat{c}k}), E(f_{cat}^{\hat{c}k}))} \tag{19}$$

Due to the high computational complexity of softmax in the encrypted scenario, we did not perform the softmax operation.

## A.2 SUPPLEMENTARY EXAMPLES OF MINDLOC VISUALIZATION RESULTS

The following figure illustrates the visualization results of the MindLoc model, showcasing how it processes and represents spatial information derived from the NSD dataset. Each image highlights different aspects of the model's performance, emphasizing its capabilities in visual localization tasks.

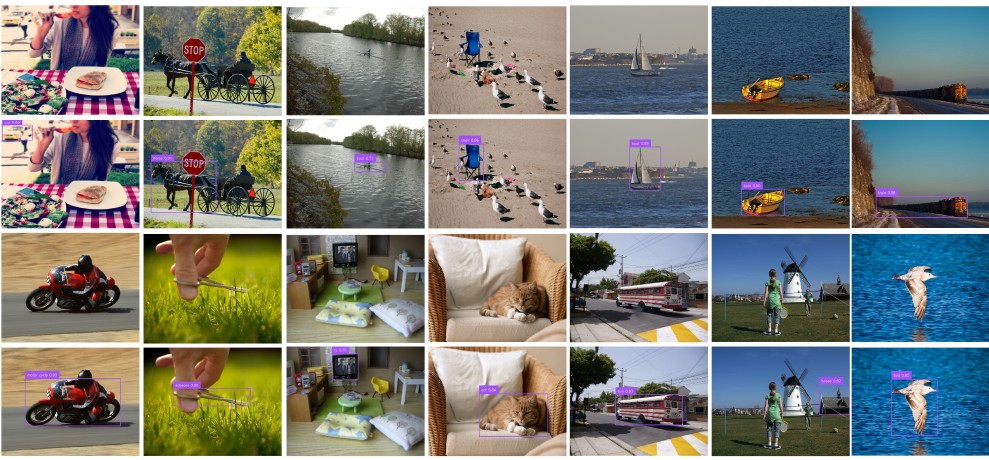

Figure 4: The visualization of the MindLoc. These images are sourced from the NSD dataset.

## A.3 ALGORITHM FOR OPTIMAL LOSS MINIMIZATION

We introduce Alg. 2, which minimizes the total loss matrix by balancing classification and IoU losses to ensure a globally optimal solution for object detection.

---

**Algorithm 2** Hungarian Algorithm for Maximum Matching Problem

---

1: **Input:** IoU matrix $M \in \mathbb{R}^{m \times n}$
2: **Output:** Optimal matching $(\text{row\_ind}, \text{col\_ind})$
3: Initialize $M[i][j] = 0, \quad \forall i \in [0, m-1], j \in [0, n-1]$
4: Compute IoU values: $M[i][j] = \text{IoU}(p_i, g_j), \quad \forall i \in [0, m-1], j \in [0, n-1]$
5: **Row reduction:** $M'[i,j] \leftarrow M[i,j] - \min(M[i,:])$
6: **Column reduction:** $M''[i,j] \leftarrow M'[i,j] - \min(M'[:,j])$
7: Mark zero elements and check if all zeros can be covered by lines
8: **if** it is possible to cover all zeros with $m$ lines **then**
9:     Matching is complete
10: **else**
11:     Find the smallest uncovered element $\delta$
12:     Adjust uncovered elements: $M'''[i,j] \leftarrow M''[i,j] \pm \delta$
13: **end if**
14: Repeat marking zero elements and adjusting the matrix until all zeros are covered by $m$ lines
15: Return the matched row and column indices $(\text{row\_ind}, \text{col\_ind})$

---

## A.4 FURTHER DESCRIPTION OF TAB. 1

Since the NSD dataset is based on COCO, its images contain multiple objects, and the target object may not always be the subject of the image. In contrast, the GOD dataset is based on ImageNet, where each image is labeled according to its main subject category. Therefore, using fMRI data from the GOD dataset is more appropriate for object localization tasks. However, since baselines such as UMBRAE cannot directly process inputs from the GOD dataset, we only present results based on the NSD dataset in our tables.

