# OpenReview forum: "MindLoc: A Secure Brain-Based System for Object Localization"
_ICLR.cc/2025/Conference — ICLR 2025 Conference Withdrawn Submission_

### Official Review · Reviewer_6ey5 · 2024-10-20

**Soundness:** 2
**Presentation:** 1
**Contribution:** 2
**Rating:** 3
**Confidence:** 4

**Summary:**

This paper propose MindLoc. MindLoc is a lightweight, cross-subject brain-based object localization model. MindLoc introduces three loss functions by aligning fMRI signals with image, caption, and category embeddings. MindLoc also adopts encryption module.

**Strengths:**

1. The addressed problem is interesting and novel.
2. The performance of MindLoc is good.

**Weaknesses:**

1. It seems that the baseline performance of brain-grounding is quite poor. The authors may need to provide a reason to explain such a significant performance difference to show why MindLoc outperforms existing baselines.
2. Although the performance of MindLoc is good among all brain-grounding methods, it seems that MindLoc still underperforms text-based methods. These experimental results are inconsistent with the red apple cases provided by the authors. The cases aim to show that the brain can provide additional information based on the text. Therefore, the authors should study on models that combine both types of information.
3. The design of the proposed method lacks theoretical foundations or experimental validation, especially in the design of loss functions.
4. Lack of experiments to show the effectiveness of the encryption module.

Minor problem:
Add citation for this sentence: However, prior work has predominantly focused on traditional modalities for localization tasks, while recent efforts have begun to explore direct localization through EEG signals, albeit with significantly lower accuracy compared to traditional models.

I think this paper is below the acceptance level for the above concerns, but I am happy to see the author's feedback and change my attitude.

**Questions:**

1. It looks strange to use the same fMRI embedding to align with different CLIP embeddings with different functions. How do you select these loss functions? Are there any theoretical foundations? I suspect a more common approach is using the same loss functions but an additional learnable mapping layer to align with different modalities.
2. The case in the introduction is a bit confusing, how do you know that red apple is in the user's mind? I suspect in the dataset collection procedure, the user's attention may be on the green apple?
3. Is there any novel design in applying the Paillier encryption scheme to fMRI modeling?
4. Why do you align fMRI-Img with L1 loss and category with sim loss?
5. In equations (4) and (5), what's the meaning of i and j as suffixes for S?
6. The caption of Figure 1 was confusing before I read the method part regarding the CS.
7. Are there any experiments to show the effectiveness of the encryption module?

---

### Official Review · Reviewer_HHcD · 2024-10-25

**Soundness:** 1
**Presentation:** 2
**Contribution:** 1
**Rating:** 3
**Confidence:** 4

**Summary:**

I am really not sure what to make of this paper. It has the appearance of being very fluid and well-written, but I’m constantly bumping into issues that completely befuddle me. The writing style convinces me I might need to think a little harder at times but then there is just no way around some of the issues of the paper, no matter how good the writing is. I’m not sure where to start or even to sum up the paper because it feels so disjointed and I can’t quite put together the logic myself. The paper offers a method to use fMRI data to do brain-based object localisation and offers a cryptography-based encryption scheme to determine object localisation in specific images using two popular datasets: NSD and GOD.

**Strengths:**

The writing style/flow is nice, the figures are good and more-or-less the structure of the paper is how I would have expected it to be.

**Weaknesses:**

First of all, the elephant in the room is the cryptography element of this paper. It seems totally disconnected from anything I would expect in a paper talking about this issue. I absolutely cannot wrap my head around it and the argumentation for using it is very tangential at best. While with structural MRIs, we often do “de-facing” before releasing the image because facial elements can be reconstructed, there hasn’t been (as far as I’m aware) even the suggested hint of participant identification via fMRI signals, so I don’t know what problem this approach solves. Does that automatically discount the utility of thinking about these ideas? No, I guess not (and that’s what I struggled with a bit). The argumentation, however, does not stack up. The authors claim we need encrypted fMRI data to “relieve the psychological burden on the participants” and that this would somehow benefit future data acquisition of similar data (but they’re using NSD??). If you are going to put effort into the encryption part, you need to make a better argument for it than this because I still just find this quite a wild idea that has little basis in reality or necessity. The claim in the paper is that this is a “major barrier to data acquisition” but this is just not my experience at all and with many years in cognitive neuroimaging, I’ve also not heard of this problem. Sure, there is a bit of trepidation about mind-reading technology in the future, but it’s already been shown how easy it is to disrupt those decoding methods (those relevant papers are not cited in this submission).

The introduction and background captures a broad theme of papers in the relevant theme of doing recent image reconstruction, particularly popular using the specific dataset. However, the background is extremely superficial and doesn’t account for the level of detail I would have expected to support the working hypothesis of the paper. This paper makes a huge assumption that “AI research” is specifically language / LLM-based and it primarily revolves around capturing human intent, which I strongly disagree with as a blanket statement. The authors should have carved out a better introduction to place their proposed work in.

The authors appear to be referring back to their own work as the “traditional approach” to brain-based object localisation in a pretty obvious way, so I wasn’t able to see that they were building on accepted work by other research groups and absence of this idea as a common theme that anyone else is thinking about, while not bad in and of itself, combined with all my other issues, makes me quite unsure what to make of this paper).

Figures are introduced with acronyms I’m unfamiliar with, without being explained, causing me to need to jump around looking for answers and losing the thread of the story. Referencing is not formatted correctly (you need to replace most of the citations with `/citep{}` to capture the parentheses). The references themselves also seem a bit off. Why are you citing a blog post in 2019 as a reference to ChatGPT, for example? ChatGPT wasn’t released until 2022.  Then some citations in 2023 have been given claiming to summarise work from 2024. It just doesn’t add up (specifically referring to lines 463-464 here). Additionally, multiple references in the bibliography for the same paper (CLIP). Some figures are just showing images with and without bounding boxes and single-line captions inform me that this is a clear demonstration of how MindLoc works as a system. Please revisit this as it's not clear to me at all what is going on with these figures.

My biggest issue, however, is that while the figures and text sound very plausible, neither NSD or GOD datasets set the participants the task of object localisation. They are presented with images for a short period of time and we don’t know what specific objects within those images were being attended. Given the same image with multiple objects, it's exactly the same fMRI data if you wanted to attend to every different object in an image (if the participants even saw all of them). The duration of the stimuli on screen was not enough time to richly capture a full understanding and so I don’t know what the source of brain data is that the authors claim is being captured here in order to boost object localisation. The authors seem to be capturing something, but the given analyses don’t make it easy to discern what’s happening but leave plenty of room for potential confounds to creep in.

Additionally, we're somehow supposed to expect that this is a big jump forward compared to standard text-based approaches. FMRI is extremely expensive and tricky to process and doesn't work in any standard real-time setting that would support the authors ideas of utility with their approach. There doesn't seem to be a sense of awareness of this issue in the text.

All these little issues, plus the confusion and lack of clarity throughout the paper cause me serious concerns about lending my support towards recommending that this paper be accepted.

**Questions:**

Please see the above weaknesses section. I am happy to be convinced by author responses and I fully state that I will keep an open mind about the responses to my critical points of assessment.

I am happy to ignore the paper's contributions regarding encryption and the whole section on cryptography because I don't think there even exists a strong argument for its utility / necessity. I am willing to assess the paper more on its merit of providing a useful brain signal to do object localisation, but with the datasets analysed and the description of the analysis performed, I want to ask the authors if they can better account for how fMRI signals to short-duration static images can be modelled in such a way to guide an object localisation system to differentially focus/attend on multiple objects in an image using the same brain data. That is the mechanism that I think is missing to establish that this entire paper even makes sense, but I wasn't convinced by the experimental description.

**Details Of Ethics Concerns:**

N/A. Uses publically available dataset for analysis.

---

### Official Review · Reviewer_vDHa · 2024-11-04

**Soundness:** 1
**Presentation:** 1
**Contribution:** 1
**Rating:** 1
**Confidence:** 2

**Summary:**

- This paper introduces a model that can localize objects in images using fMRI brain responses.
- An encryption module is included for privacy considerations.
- The system is compared to other brain-based localization models.

**Strengths:**

- High accuracies are reported.

**Weaknesses:**

- The methods section 3.1 is very confusing. Many loss functions are introduced with undefined terms and missing context.
- An encryption module is included, however NSD and GOD datasets are already anonymized. The inclusion of encryption seems out of place in this paper.
- The other methods compared to in table 1 are barely described and are missing citations in the text. There is also no visual comparison to these methods.

**Questions:**

How is the ground-truth class determined for the MS-COCO stimulus images that were used in NSD?

---

### Note · Authors · 2024-11-12

I have read and agree with the venue's withdrawal policy on behalf of myself and my co-authors.